# Alcohol Consumption, Loneliness, Quality of Life, Social Media Usage and General Anxiety before and during the COVID-19 Pandemic in Singapore

**DOI:** 10.3390/ijerph19095636

**Published:** 2022-05-05

**Authors:** Mengieng Ung, Kalista Yearn Yee Wan, Shi Yu Liu, Ying Jie Choo, Nathaniel Shan Wei Liew, Zhexuan Azure Shang, Sophie Su Hui Khoo, Wei Xuan Tay, Ruixi Lin, Siyan Yi

**Affiliations:** 1Saw Swee Hock School of Public Health, National University of Singapore and National University Health System, Singapore 117549, Singapore; mung@nus.edu.sg (M.U.); ruixi-l@nus.edu.sg (R.L.); 2Yong Loo Lin School of Medicine, National University of Singapore, Singapore 117597, Singapore; e0196612@u.nus.edu (K.Y.Y.W.); e0326089@u.nus.edu (S.Y.L.); e0345826@u.nus.edu (Y.J.C.); nathanielliew@u.nus.edu (N.S.W.L.); e0326077@u.nus.edu (Z.A.S.); sophiekhoo@u.nus.edu (S.S.H.K.); e0326057@u.nus.edu (W.X.T.); 3KHANA Center for Population Health Research, Phnom Penh 12301, Cambodia; 4Center for Global Health Research, Touro University California, Vallejo, CA 94592, USA

**Keywords:** COVID-19, pandemic, GAD-7, mental health, quality of life, Asia

## Abstract

This cross-sectional study aims to identify factors associated with anxiety levels of adults living in Singapore before and during the COVID-19 pandemic. Data were collected using a web-based survey conducted from July to November 2020, accruing 264 eligible participants. Ordered logistic regression was used to assess factors associated with Generalized Anxiety Disorder-7 (GAD-7), ranked as minimal (0–4), mild (5–9), moderate (10–14), and severe (15–21) before and during the pandemic. About 74% of participants were female, 50% were aged 25–34, and 50% were married. The GAD-7 level went up from the pre-pandemic for moderate (12.5% to 16%) and severe GAD (2% to 11%). Alcohol consumption (AOR 1.79, 95% CI 1.04–3.06), loneliness (AOR 1.28, 95% CI 1.05–1.54), and difficulty in switching off social media (AOR 2.21, 95% CI 1.29–3.79) predicted increased GAD-7 levels. The quality of life (AOR 0.84, 95% CI 0.79–0.90) was significantly associated with decreased GAD-7 levels. The results heighten the awareness that early initiation of mental health support is crucial for the population in addition to the various financial support measures provided by the government as they are adapting to live with the COVID-19 pandemic.

## 1. Introduction

Singapore was among the first few countries in the region and the world to detect COVID-19 cases outside China. The first Singaporean case of COVID-19 infection was reported on 23 January 2020. With increasing numbers of local transmission, a series of strict island-wide measures termed “Circuit Breaker’’ were put in place on 7 April 2020, to reduce local transmission. Such measures included shifting to work from home for most non-essential workplaces, introducing home-based learning, tightening entry restrictions to public spaces, and banning in-person social gatherings among individuals from different households [1]. These measures remained in place for two months, from April to June 2020, following which a ‘new normal’ for Singaporeans was borne with variations of these restrictions persisting into Singapore’s future. People in Singapore had to become used to the sudden changes in life, such as participating in social distancing measures and restrictions on social activities, such as dining out and working and studying from home.

Research has shown that tremendous changes brought about by the COVID-19 pandemic have culminated in unprecedented mental health effects on individuals worldwide [2,3]. A study conducted in Kuwait during the last few days of the lockdown in May 2020 found that increased social media usage was a significant risk factor for anxiety and depression [4]. Another study conducted during the pandemic in Australia highlighted the association between reduced physical activity and increased alcohol consumption with higher depression, anxiety, and stress symptoms [5]. The COVID-19 safety restrictions have correlated with increased reported adverse psychological effects, including post-traumatic stress symptoms [6], a high prevalence of generalized anxiety symptoms, depressive symptoms, psychological distress, and COVID-19 related fear [7].

Of these psychological effects, anxiety seems to be the predominant issue that has seen a marked increase in the incidence and prevalence since the beginning of this pandemic. A study by Bäuerle and colleagues in Germany during the COVID-19 pandemic from March to May 2020 suggested a 44.9% prevalence of elevated general anxiety symptoms [8]. Similarly, a study conducted in the United Arab Emirates found that anxiety levels, measured by the Generalized Anxiety Disorder 7-item Scale (GAD-7), were notably higher at 55.7% than those reported in pre-pandemic studies [9,10]. These findings highlight the enormous impact the COVID-19 pandemic has on mental health—an observed trend in other countries that could apply to the Singapore population. In Singapore, a survey conducted one year after the “Circuit Breaker’’ found that 61% of the 1000 respondents reported that they socialized less frequently with those outside their immediate family, 41% reported that their social circles outside of their immediate family have shrunk over the past year, and 36% reported that their mental health has worsened [11].

Several factors are associated with anxiety levels. The inverse relationship between quality of life (QoL) and anxiety has been well established in psychology work [12,13]. Likewise, a longitudinal study among an older Irish population in 2019 suggested that loneliness is predictive of anxiety levels [14]. Alcohol dependence was associated with increased persistence of anxiety disorders [15] among 1369 adults in the Netherlands. The impact of social media on mental health has also been widely documented in the literature. For instance, a 2017 cross-sectional study of people aged 18–22 years old in the United States found that increased time spent on social media was associated with more significant symptoms of dispositional anxiety and a high GAD-7 score [16]. Furthermore, a systematic review on social media use and anxiety suggested a positive correlation between the two variables [17,18]. The impact of social media on anxiety became more pronounced during the COVID-19 pandemic when people turned to social media to be informed of the virus update. Another systematic review suggested that spending time on social media is associated with heightened anxiety levels [19]. While studies have shown an individual relationship with anxiety, a combined effect of QoL, loneliness, alcohol consumption, and social media usage on anxiety remains unclear. As the COVID-19 pandemic persists, mental wellbeing remains an emerging and pertinent societal issue. A clearer understanding of how mental health has been affected by the pandemic in Singapore is imperative to promote awareness and targeted interventions to better address the contemporary mental health issue.

Literature shows that research focusing on the mental health of the general Singapore population during the COVID-19 pandemic is sparse. Nevertheless, the pandemic’s impact on specific communities in Singapore, such as migrant workers [20] and healthcare workers [21,22] has been documented. Hence, this study aimed to delineate the impact of the COVID-19 pandemic on the mental health of Singapore residents, focusing specifically on anxiety levels. We assessed how different factors—demographics, socioeconomic status, QoL, loneliness, alcohol consumption, and social media usage—impact anxiety levels among Singapore residents during the pre-pandemic (December 2019) and the pandemic (at the time of the survey).

## 2. Materials and Methods

### 2.1. Study Design and Participants Recruitment

We conducted this cross-sectional study in Singapore as part of a larger multi-country consortium online survey on “Personal and Family Coping with COVID-19 Crisis: A Multi-Country Online Survey”, hosted on ICP-COVID (International Citizen Project COVID-19). Convenience and snowball sampling approaches were used to select voluntary participants for this study from July to November 2020. A self-administered questionnaire was uploaded online and disseminated through professional and social circles via communication and social media platforms, such as emails, WhatsApp, Telegram, and Messenger. Potential participants were asked to join the study and forward the flyer to their circles. Participation was restricted to adults aged 21 years and above and currently residing in Singapore.

### 2.2. Questionnaire Design

We used psychometrically validated instruments to develop the questionnaire, such as GAD-7, WHO-5 Well-Being Index, and the three-item Loneliness Scale. The survey utilized the Likert-like scale and generic ordered-category system, conferring ease of quantitative analyses. The questionnaire featured the following segments: sociodemographic characteristics and impact of the COVID-19 pandemic on wellbeing, anxiety, loneliness, alcohol consumption, and media and social media consumption patterns.

### 2.3. Variables and Measurements

#### 2.3.1. Generalized Anxiety Disorder

Outcome variables were the GAD before the pandemic in December 2019 (GAD-Before) and the GAD during the pandemic in July–November 2020 (GAD-COVID). Questions on participants’ reported experiences before the pandemic were asked retrospectively. The GAD-7 scale [23] asked participants how often, during the last two weeks, they had been bothered by: (1) feeling nervous, anxious, or on edge, (2) not being able to stop or control worrying, (3) worrying too much about different things, (4) trouble relaxing, (5) being so restless that it is hard to sit still, (6) becoming easily annoyed or irritable, and (7) feeling afraid as if something awful might happen. Each question has answer options of 0 = not at all, 1 = several days, 2 = more than half of the days, and 3 = nearly every day. Consequently, the GAD-7 scores ranged from 0 to 21, with scores 0–4, 5–9, 10–14, and 15–21 representing minimal, mild, moderate, and severe anxiety symptom levels, respectively [24]. The decision to use a cut-off of 8 was informed by a meta-analysis, where some experts have recommended using this cut-off to optimize sensitivity without compromising specificity [25]. The internal consistency of the scale items was assessed using item correlations and Cronbach’s alpha. The Cronbach’s alpha of GAD-Before (α = 0.88) and GAD-COVID (α = 0.90) in this study were well above the recommended threshold of 0.70.

#### 2.3.2. Sociodemographic Characteristics, Alcohol Use, and Social Media Use

The predictor variables included sociodemographic characteristics, including sex (female, male), age group (21–24, 25–34, 35–44, 45–54, 55–64, 65 and older), marital status (single, married, unmarried cohabitation), education (post-secondary or below, undergraduate, graduate), and wealth quintile (lower income, higher income). Exact income brackets were not specified in the survey to assess participants’ self-perception of whether they belonged to a low- or high-income bracket. Nevertheless, for a rough reference, according to the Department of Statistics Singapore in 2020, the median monthly household income per household member was SGD 2886 [26]. We also asked the participants about alcohol consumption before the COVID-19 pandemic (no, yes) and during the COVID-19 pandemic (no, yes), and whether they found it difficult for them to switch off social media in the past two weeks (no, yes).

#### 2.3.3. Quality of Life

The WHO (Five) Well-Being Index, 1998 version [27] was used to measure the QoL. The WHO-5 index asks participants if they have felt: (1) cheerful in good spirits, (2) calm and relaxed, (3) active and vigorous, (4) fresh and rested upon waking up, and (5) their life has been filled with things that interest them. Each question has answer options of: 0 = at no time, 1 = some of the time, 2 = less than half the time, 3 = more than half the time, 4 = most of the time, and 5 = all the time. Consequently, the QoL scores ranged from 0 to 25. The questions were asked both before the COVID-19 pandemic (QoL-Before) and during the COVID-19 pandemic (QoL-COVID). The Cronbach’s alpha of QoL-Before (α = 0.85) and QoL-COVID (α = 0.86) were well above the recommended threshold of 0.70.

The Short Scale for Measuring Loneliness in Large Surveys [28] using the 3-item index was used to measure participants’ loneliness before and during the COVID-19 pandemic. The index asked participants, in the past two weeks, how often they felt: (1) lack of companionship, (2) left out, and (3) isolated from others. Each question has answer options of: 1 = hardly ever, 2 = some of the time, and 3 = often, making the loneliness scores range from 3 to 9. The Cronbach’s alpha of Loneliness-Before (α = 0.77) and Loneliness-COVID (α = 0.78) were above the recommended threshold of 0.70. For the last predictor variable, we asked participants how difficult it was for them to switch off from media and social media (easy, difficult).

### 2.4. Data Analyses

We conducted descriptive, bivariate, and multiple analyses using Stata (StataCorp LP, version 15 (StataCorp LLC, College Station, TX, USA)). For descriptive analyses, absolute and relative frequency (percentages) were calculated to describe categorical variables. Arithmetic mean and standard deviation (SD) were used for continuous variables. For both bivariate and multiple analyses, ordered logistic regression was used to accommodate the nature of the outcome variables. GAD-Before and GAD-COVID ranked minimal, mild, moderate, and severe anxiety symptom levels. We constructed two models for multiple regression analyses. Model I included sociodemographic factors, and Model II included all other covariates associated with alcohol consumption, QoL, loneliness, and difficulty switching off media and social media. We calculated odds ratios (OR) in bivariate analyses and adjusted odds ratios (AOR) in multiple logistic regression analyses with confidence intervals (CI). In all analyses, *p*-values < 0.05 were considered statistically significant.

### 2.5. Ethical Consideration

This study was approved by the Institutional Review Board (IRB) of the National University of Singapore (reference: NUS-IRB-2020-166). Participation in the survey was voluntary. Participants had to read and acknowledge the study’s information provided at the beginning before proceeding with the survey. No personal identifiers were linked to the questionnaire, ensuring the participants’ anonymity and confidentiality. Data were coded, made accessible only to the principal investigators, and stored in a password-protected computer.

## 3. Results

### 3.1. Sociodemographic Characteristics, Alcohol Use, and Social Media Use

A total of 264 eligible participants completed the survey. Table 1 summarizes an overview of the variables included in the analyses. Around 56% (95% CI: 49.97–61.97) of respondents reported minimal, 30% (95% CI: 24.32–35.37) reported mild, 13% (95% CI: 9.00–17.09) reported moderate, and 2% (95% CI: 0.78–4.50) reported severe GAD before the pandemic. The proportions of respondents with minimal, mild, moderate, and severe GAD during the COVID-19 pandemic were 44.32% (95% CI: 38.39–50.40), 29.17% (95% CI: 23.97–34.97), 15.91% (95% CI: 11.95–20.87), and 10.61% (95%CI: 7.41–14.96), respectively.

Close to 74% of participants were female. About half (50.8%) were aged 25 to 34 years old. Most of the participants (87%) were employed. Regarding education level, 45% received undergraduate education, 37% had a graduate degree, and around 17% completed post-secondary education. Approximately 56% reported being in the lower-income bracket.

More than half of respondents reported drinking alcohol before the COVID-19 pandemic, while 49% reported doing so during the pandemic. Mean QoL scores decreased from 13.9 (SD 4.55) before COVID-19 to 11.3 (SD 4.63) during COVID-19. In contrast, mean loneliness scores increased from 4.58 (SD 1.46) before the COVID-19 pandemic to 5.20 (SD 1.74) during the pandemic. Lastly, almost 40% of the participants reported difficulty switching off social media during the COVID-19 pandemic.

### 3.2. Factors Associated with Anxiety Symptoms

Table 2 shows factors associated with anxiety symptom levels (GAD-7) before the COVID-19 pandemic in bivariate and multiple logistic regression analyses. Married individuals and those in the high-income quintile were significantly less likely to report increased anxiety symptoms (OR 0.57, 95% CI 0.35–0.92 and OR 0.61, 95% CI 0.37–0.98, respectively). The relationship remained unchanged after controlling for other sociodemographic variables in the multiple regression analysis (Model I).

Participants who reported having consumed alcohol in the past week (OR 1.70, 95% CI 1.05–2.74), found it difficult to switch off social media (OR 1.98, 95% CI 1.21–3.22), and those who scored higher on the loneliness scale (OR 1.47, 95% CI 1.24–1.73) were significantly more likely to report increased anxiety symptoms in the bivariate analyses. On the contrary, those who scored higher on the QoL scale were less likely (OR 0.84, 95% CI 0.79–0.89) to report increased anxiety symptoms before the COVID-19 pandemic (Table 2).

After adjusting for all other covariates in Model II, participants who reported having consumed alcohol in the past week (AOR 1.79, 95% CI 1.04–3.06), scored higher on the loneliness scale (AOR 1.28, 95% CI 1.05–1.54), and had difficulty in switching off social media (AOR 2.12, 95% CI 1.23–3.65), had higher odds of reporting higher levels of GAD before the COVID-19 pandemic. Lastly, respondents with a higher QoL score had lower odds (AOR 0.84, 95% CI 0.79–0.90) of having increased anxiety symptoms (Table 2).

Results of bivariate and multiple ordered logistic regression analyses on GAD during the pandemic are presented in Table 3. In bivariate analyses, participants who held a graduate degree (OR 0.46, 95% CI 0.24–0.89), were in the high-income bracket (OR 0.51, 95% CI 0.32–0.82), and reported a higher QoL score (OR 0.74, 95% CI 0.70–0.79) were significantly less likely to report increased anxiety symptoms. Conversely, participants who reported higher loneliness scores (OR 1.74, 95% CI 1.50–2.02) and those who found it difficult to switch off social media (OR 2.35, 95% CI 1.47–3.76) were significantly more likely to report increased anxiety symptoms.

In multiple logistic regression analyses, participants in the high-income bracket were significantly less likely (AOR 0.60, 95% CI 0.37–0.99) to report increased anxiety symptoms during the COVID-19 pandemic after adjusting for other sociodemographic variables (Model I). In Model II, the odds of having increased anxiety symptoms were significantly lower among participants who reported a higher QoL score (AOR 0.77, 95% CI 0.72–0.82). In contrast, the odds of reporting increased anxiety symptoms were significantly higher among those who reported higher loneliness scores (AOR 1.45, 95% CI 1.23–1.72) and those who found it difficult to switch off social media (AOR 2.21, 95% CI 1.29–3.79).

## 4. Discussion

The COVID-19 pandemic has led to a spark in unprecedented mental health effects among individuals globally due to the unpredictability of the pandemic and living with the new normal, which is congruent with existing literature. A recent study by Wang et al. in China during the pandemic, revealed that about a third of 1210 respondents reported moderate to severe anxiety, and 53% of the respondents rated the overall psychological impact of the outbreak to be moderate to severe [29]. Of note, GAD levels among the general population in Singapore remarkably increased through this pandemic. Our study found that alcohol consumption, QoL, loneliness, and the difficulty in switching off social media were significantly related to the GAD levels in adult residents in Singapore.

Alcohol consumption was significantly associated with the GAD level of the general population in this study before the COVID-19 pandemic. The finding is corroborated with the literature suggesting that alcohol-use disorders [30] are correlated with increased anxiety, and the higher level of anxiety and alcohol use disorders are often comorbid [31,32]. A study in the United States reported an increase (29%) in alcohol consumption since the COVID-19 pandemic started [33]. In addition, it was noted that those with anxiety or depressive symptoms had significantly higher odds of reporting increased alcohol use than those without symptoms [34].

However, in our study, alcohol consumption did not increase during the COVID-19 pandemic and was not associated with GAD levels. This might be because fewer people reported consuming alcohol during the pandemic, likely due to financial concerns from job insecurity and the imposed restrictions on social gathering activities and venues. A Labor Market Report released by the Ministry of Manpower in the second quarter of 2020 reported that total employment saw the steepest quarterly contraction [35]. The retrenchments more than doubled from the previous quarter [35]. Against the backdrop of the COVID-19 pandemic, job insecurity faced by those living in Singapore would be a plausible contributing factor to financial concerns and reduced expenditure on activities such as alcohol drinking. Additionally, from March 2020, entertainment venues such as pubs, bars, and discotheques were shuttered due to concerns of high-risk transmission of COVID-19. Pilot reopening measures were implemented for a small portion of the nightlife business in November 2020, but not without enforcement measures [36]—restrictions of group sizes, social distancing, and banning of live performances—which compromises the entire social drinking experience.

A higher score of QoL was inversely associated with a higher level of GAD scores, which is intuitive. Though there exists minimal literature yet drawing this conclusion, the higher anxiety level correlated with poorer QoL has been well demonstrated in numerous studies, such as those by Bourland et al. [37], Barrera et al. [38], Henning et al. [39], and Olatunji et al. [13]. Similarly, in Spain, González-Blanch et al. found that the intensity of the GAD predicted poorer psychological and physical QoL domains [40].

In our study, social media usage was related to increased anxiety symptoms before and during the COVID-19 pandemic. This finding is consistent with previous studies conducted before the advent of COVID-19 on elderly populations in the United States [41,42] and a study conducted during COVID-19 in the United Kingdom [43]. Mobile device use is ubiquitous in Singapore, and the internet has become increasingly accessible to the general population, with the rise in internet users from 36% in 2000 to 87% in 2018 [44]. This increase in internet usage sparked the advent of online communication and social media usage. As shown in a 2012 report, 90% of the nearly 3 million internet users in Singapore visited social media sites [45]. The “new normal” brought about by the COVID-19 pandemic has resulted in a further increase in social media usage [46]. People have turned to social media to feel connected to others when social distancing measures are implemented, limiting physical interaction. A recent study found that 59.7% of tertiary students in Singapore reported increased use of Instagram during the COVID-19 pandemic period [47]. This social media use was associated with increased anxiety symptoms in our study. It has been postulated that social media exacerbates poor mental health issues [48]. Furthermore, a recent study in China found that increased anxiety and poor mental health are positively associated with increased social media usage during the COVID-19 pandemic [49], which concurs with our study. The overspreading of fake news on the social media platform regarding the COVID-19 measures and preventions might further trigger the anxiety level of the general population.

Loneliness was associated with increased GAD levels in this study. This finding is consistent with the literature. For instance, the subjectively weak social integration and low social relations that lead to loneliness were identified as risk factors associated with anxiety among the university students in France who experienced COVID-19 quarantine measures [50]. In the United States, loneliness was significantly associated with clinical anxiety levels among young adults [51] and the general population [52]. However, a bi-directional relationship between anxiety and loneliness is worth mentioning. For instance, a global systematic review found moderate associations between GAD and loneliness among adolescents, young adults, and children [53].

Our study has a few limitations. Firstly, data collection was conducted using an online survey, mainly disseminated via emails, WhatsApp, Telegram, and other online platforms within the study team’s personal and professional circles. This may lead to selection bias, as our demographics could have been skewed towards younger and more educated groups. In particular, the elderly who are less technologically inclined might not participate in the survey. Secondly, this cross-sectional study cannot determine causality in the findings. Thirdly, the self-administering nature of the survey could have led to recall and social desirability biases. Finally, the study’s small sample size may have accounted for the inability to generalize to the larger population.

Nevertheless, this study offers insights into the GAD levels of the Singaporean population before and during the COVID-19 pandemic. To the authors’ knowledge, this is the first study looking into the effect of the pandemic on anxiety levels among the general population in Singapore, assessing how different factors correlate with anxiety levels before and during the pandemic. Findings from this study contribute to the broader understanding of the pandemic’s impacts beyond specific target groups, such as healthcare workers, patients with cancer, and individuals with family members infected by COVID-19.

## 5. Conclusions

The COVID-19 pandemic has unprecedented impacts on the Singapore population with purported increased anxiety levels. We found that the increased GAD levels before and during the COVID-19 pandemic are significantly associated with the QoL, loneliness, and difficulty switching off social media. However, alcohol consumption is significantly related to the GAD level only before the pandemic. While the Singapore government implements various measures supporting Singaporeans financially, such as the COVID-19 Recovery Fund and pay-outs from the Courage Fund, mental health is paramount. Intervention strategies should be established in community clubs, schools, and workplaces to enhance socialization, reduce loneliness, and raise mental health awareness and support in this challenging period. The strategies may include arranging and publicizing in-person bonding activities following recommended safe distancing precautions and counseling services for people living with stress and loneliness. With greater awareness and adequate infrastructure in place, the impact of the pandemic on mental health can hopefully be kept to a minimum. 

## Figures and Tables

**Table 1 ijerph-19-05636-t001:** Descriptive characteristics of the study participants (n = 264).

Variables	Frequency (%)
GAD-7 Before the COVID-19 pandemic	
Minimal	148 (56.06)
Mild	78 (29.55)
Moderate	33 (12.5)
Severe	5 (1.89)
GAD-7 During the COVID-19 pandemic	
Minimal	117 (44.32)
Mild	77 (29.17)
Moderate	42 (15.91)
Severe	28 (10.61)
Female	195 (73.86)
Age	
21–24	22 (8.33)
25–34	134 (50.76)
35–44	53 (20.08)
45–54	35 (13.26)
55–64	13 (4.92)
65+	7 (2.65)
Married	134 (50.76)
Education	
Post-secondary or below	46 (17.42)
Undergraduate	120 (45.45)
Graduate	98 (37.12)
Unemployed	34 (12.88)
Lower income	148 (56.06)
Drinking alcohol (before the COVID-19 pandemic)	123 (46.59)
Drinking alcohol (during the COVID-19 pandemic)	133 (50.38)
Quality of life (before the COVID-19 pandemic), mean (SD)	13.86 (4.55)
Quality of life (during the COVID-19 pandemic), mean (SD)	11.33 (4.63)
Loneliness (before the COVID-19 pandemic), mean (SD)	4.58 (1.46)
Loneliness (during the COVID-19 pandemic), mean (SD)	5.20 (1.74)
Difficult to switch off social media in the past two weeks	94 (35.61)

GAD, Generalized Anxiety Disorder 7-items; SD, standard deviation.

**Table 2 ijerph-19-05636-t002:** Factors associated with anxiety symptom levels (GAD-7) before the COVID-19 pandemic in ordered logistic regression analysis (n = 264).

Predictor Variables	OR (95% CI)	AOR (95% CI)	
		Model I	Model II
Sex			
Female	1	1	1
Male	0.68 (0.39–1.18)	0.75 (0.42–1.34)	0.96 (0.52–1.77)
Age			
21–24	1	1	1
25–34	0.61 (0.26–1.40)	0.68 (0.25–1.83)	0.99 (0.33–2.96)
35–44	0.45 (0.17–1.15)	0.56 (0.18–1.72)	0.55 (0.15–1.95)
45–54	0.54 (0.19–1.50)	0.69 (0.20–2.39)	1.20 (0.30–4.73)
55–64	0.22 (0.05–1.01)	0.27 (0.05–1.45)	0.54 (0.09–3.26)
65+	0.43 (0.08–2.17)	0.82 (0.14–4.68)	1.27 (0.19–8.30)
Marital status			
Unmarried	1	1	1
Married	0.57 (0.35–0.92) *	0.54 (0.30–0.96) *	0.75 (0.40–1.40)
Education			
Post-secondary or below	1	1	1
Undergraduate	0.75 (0.38–1.47)	0.60 (0.28–1.30)	0.50 (0.22–1.12)
Graduate	0.83 (0.42–1.65)	0.88 (0.39–1.97)	0.76 (0.32–1.78)
Employment status			
Unemployed	1	1	1
Employed	1.29 (0.64–2.60)	2.19 (0.96–5.01)	1.96 (0.80–4.80)
Income quintile			
Low income	1	1	1
Higher income	0.61 (0.37–0.98) *	0.56 (0.33–0.96) *	0.66 (0.36–1.19)
Alcohol drinking in the past week			
No	1		1
Yes	1.70 (1.05–2.74) *		1.79 (1.04–3.06) *
Level of quality of life			
Lower	1		1
Higher	0.84 (0.79–0.89) ***		0.84 (0.79–0.90) ***
Level of loneliness			
Lower	1		1
Higher	1.47 (1.24–1.73) ***		1.28 (1.05–1.54) **
Difficulty in switching off social media			
No	1		1
Yes	1.98 (1.21–3.22) **		2.12 (1.23–3.65) ***

AOR, adjusted odds ratio; CI, confidence interval; GAD-7, Generalized Anxiety Disorder 7-item; OR, odds ratio. * *p* < 0.05, ** *p* < 0.01, *** *p* < 0.001.

**Table 3 ijerph-19-05636-t003:** Factors associated with anxiety symptom levels (GAD-7) during the COVID-19 pandemic (n = 264).

Predictor Variables	OR (95% CI)	AOR (95% CI)	
		Model I	Model II
Sex			
Female	1	1	1
Male	0.64 (0.38–1.07)	0.67 (0.39–1.14)	0.90 (0.50–1.62)
Age			
21–24	1	1	1
25–34	0.92 (0.42–2.01)	1.27 (0.49–3.27)	1.29 (0.43–3.86)
35–44	0.76 (0.32–1.82)	1.10 (0.39–3.10)	0.92 (0.27–3.16)
45–54	0.67 (0.26–1.76)	0.98 (0.30–3.18)	1.05 (0.28–3.95)
55–64	0.63 (0.17–2.30)	0.88 (0.21–3.74)	0.92 (0.17–4.83)
65+	0.62 (0.11–3.26)	1.15 (0.19–6.69)	0.73 (0.11–4.68)
Marital status			
Unmarried	1	1	1
Married	0.80 (0.51–1.25)	0.80 (0.47–1.36)	1.25 (0.70–2.25)
Education			
Post-secondary or below	1	1	1
Undergraduate	0.79 (0.42–1.47)	0.69 (0.34–1.39)	0.80 (0.36–1.74)
Graduate	0.46 (0.24–0.89) *	0.48 (0.22–1.01)	0.79 (0.34–1.83)
Employment status			
Unemployed			
Employed	1.22 (0.62–2.37)	1.56 (0.71–3.39)	0.88 (0.37–2.06)
Income quintile			
Low income	1	1	1
Higher income	0.51 (0.32–0.82) **	0.60 (0.37–0.99) *	0.98 (0.55–1.72)
Alcohol drinking in the past week			
No	1		1
Yes	0.87 (0.55–1.36)		0.94 (0.56–1.57)
Level of quality of life			
Lower	1		1
Higher	0.74 (0.70–0.79) ***		0.77 (0.72–0.82) ***
Level of loneliness			
Lower	1		1
Higher	1.74 (1.50–2.02) ***		1.45 (1.23–1.72) ***
Difficulty in switching off social media			
No	1	1	1
Yes	2.35 (1.47–3.76) ***		2.21 (1.29–3.79) ***

AOR, adjusted odds ratio; CI, confidence interval; GAD-7, Generalized Anxiety Disorder 7-item; OR, odds ratio. * *p* < 0.05, ** *p* < 0.01, *** *p* < 0.001.

## Data Availability

The data used in this study are available on request to the corresponding author (siyan@nus.edu.sg), subject to the approval of the National University of Singapore’s Institutional Review Board.

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
