# Peer review of "Alcohol Consumption, Loneliness, Quality of Life, Social Media Usage and General Anxiety before and during the COVID-19 Pandemic in Singapore"

_ijerph, 2022, doi:10.3390/ijerph19095636_

Round 1

Reviewer 1 Report

Dear authors:

I have read with great interest your manuscript. I think the objectives and hypothesis of this work are interesting.

I think the design is not appropriate to describe the prevalence of anxiety, as the sample size is small and the sample is not representative of the target population (selection bias). Moreover, authors do not use confidence interval for this description.

Regarding the associations between different factors and anxiety, authors should discuss the implications of the cross-sectional design of the study. This is a very important limitation that should be not only acknowledged but discussed about its implications. For instance, loneliness may cause anxiety or the other way around.

Moreover, it would be interesting to formally compare the magnitude of the association before and during COVID-19 for each exposure, for instance using a Ratio of Odds Ratios.

It may also be interesting to study those participants who actually changed from no anxiety to anxiety and see which exposure factors also changed.

I think the manuscript should be revised to highlight the results about the associations and not the description, in the introduction, the discussion and the conclusions.

Author Response

Comments

Response

Reviewer 1

I have read with great interest your manuscript. I think the objectives and hypothesis of this work are interesting.

Thank you for your comment. We appreciate you spending the time to review our manuscript.

I think the design is not appropriate to describe the prevalence of anxiety, as the sample size is small and the sample is not representative of the target population (selection bias). Moreover, the authors do not use confidence interval for this description.

We agree that the study design and small sample size did not allow us to measure the prevalence of anxiety in the population. We have avoided using the word prevalence in the paper. We have also acknowledged the small sample size as part of the limitation of this study and that we cannot generalize the findings to the larger population.

We have included confidence intervals for the proportions of participants with minimal, mild, moderate, and severe GAD before and during the COVID-19 pandemic in the first paragraph of the results section.

Regarding the associations between different factors and anxiety, authors should discuss the implications of the cross-sectional design of the study. This is a very important limitation that should be not only acknowledged but discussed about its implications. For instance, loneliness may cause anxiety or the other way around.

We have discussed the bi-directional relationship between loneliness and anxiety in the discussion section (the last third paragraph of the section).

Moreover, it would be interesting to formally compare the magnitude of the association before and during COVID-19 for each exposure, for instance using a Ratio of Odds Ratios.

We have compared the magnitude of the association of GAD before and during the pandemic; however, the outcomes were not statistically significant.

It may also be interesting to study those participants who actually changed from no anxiety to anxiety and see which exposure factors also changed.

Thank you for suggesting this interesting idea. We have captured participants who changed from a lower level of anxiety before the COVID-19 pandemic to a higher level of anxiety during the pandemic. However, the number was too small to have sufficient statistical power for the comparison.

I think the manuscript should be revised to highlight the results about the associations and not the description, in the introduction, the discussion and the conclusions.

Our manuscript highlights the results of the associations between anxiety and several factors included in the survey in the introduction, discussion, and conclusions.

Reviewer 2 Report

I have appreciated the limitations paragraph in the manuscript "Consumption, Loneliness, Qualit very welly of Life, Social Media Usage, and General Anxiety Before and During the COVID-19 Pandemic in Singapore" by Ung et al.. This paragraph describes the problems of this study very well. I agree with the authors on considering the data reported significance and the limits in the application to the general population of Singapore. It is possible to improve the clinical value by calculating the Cohen coefficient, which increases the interpretation of differences found.

Author Response

Comments Response
Reviewer 2  
I have appreciated the limitations paragraph in the manuscript "Consumption, Loneliness, Quality of Life, Social Media Usage, and General Anxiety Before and During the COVID-19 Pandemic in Singapore" by Ung et al. This paragraph describes the problems of this study very well. I agree with the authors on considering the data reported significance and the limits in the application to the general population of Singapore. It is possible to improve the clinical value by calculating the Cohen coefficient, which increases the interpretation of differences found. We appreciate your time in reviewing our manuscript and encouraging comments. In our understanding, Cohen's d is usually used to measure effect size when you have a dichotomous predictor and a continuous outcome, which is not our case. 

Round 2

Reviewer 2 Report

No comments need

Author Response

We would like to take this opportunity to thank the reviewer for sharing constructive expert insights for improving our manuscript.